# Does moderate alcohol consumption accelerate the progression of liver disease in NAFLD? A systematic review and narrative synthesis

Helen Jarvis ![ORCID],[1] Hannah O'Keefe,[1] Dawn Craig,[1] Daniel Stow ![ORCID] ,[1] Barbara Hanratty ![ORCID] ,[1] Quentin M Anstee[2]

BH and QMA are joint last authors.

¹Population Health Sciences Institute, Newcastle University, Newcastle upon Tyne, UK
²Translational and Clinical Research Institute, Newcastle University, Newcastle upon Tyne, UK

**Correspondence to**
Dr Helen Jarvis;
helen.jarvis2@ncl.ac.uk

## ABSTRACT

**Objectives** Liver disease is a leading cause of premature death, partly driven by the increasing incidence of non-alcohol-related fatty liver disease (NAFLD). Many people with a diagnosis of NAFLD drink moderate amounts of alcohol. There is limited guidance for clinicians looking to advise these patients on the effect this will have on their liver disease progression. This review synthesises the evidence on moderate alcohol consumption and its potential to predict liver disease progression in people with diagnosed NAFLD.

**Methods** A systematic review of longitudinal observational cohort studies was conducted. Databases (Medline, Embase, The Cochrane Library and ClinicalTrials. gov) were searched up to September 2020. Studies were included that reported progression of liver disease in adults with NAFLD, looking at moderate levels of alcohol consumption as the exposure of interest. Risk of bias was assessed using the Quality in Prognostic factor Studies tool.

**Results** Of 4578 unique citations, 6 met the inclusion criteria. Pooling of data was not possible due to heterogeneity and studies were analysed using narrative synthesis. Evidence suggested that any level of alcohol consumption is associated with worsening of liver outcomes in NAFLD, even for drinking within recommended limits. Well conducted population based studies estimated up to a doubling of incident liver disease outcomes in patients with NAFLD drinking at moderate levels.

**Conclusions** This review found that any level of alcohol intake in NAFLD may be harmful to liver health. Study heterogeneity in definitions of alcohol exposure as well as in outcomes limited quantitative pooling of results. Use of standardised definitions for exposure and outcomes would support future meta-analysis. Based on this synthesis of the most up to date longitudinal evidence, clinicians seeing patients with NAFLD should currently advise abstinence from alcohol.

**PROSPERO registration number** The protocol was registered with PROSPERO (#CRD42020168022).

## INTRODUCTION

Liver disease is an increasing health burden across the world, and it is now a major cause of premature (<65 years) mortality.[1 2] As premature mortality rates from many non-communicable diseases have fallen over the last 30 years, the burden of liver disease is increasing.[2 3] The most common causes of chronic liver disease in high-income countries are alcohol-related liver disease (ARLD) and metabolic-syndrome-related liver disease (or non-alcohol-related fatty liver disease—NAFLD). Chronic liver disease is often diagnosed as a result of abnormal liver blood tests or liver imaging, with a fatty liver (steatosis) progressing in some through inflammation (steatohepatitis) and stiffening (fibrosis) to scarring (cirrhosis) increasing the risk of decompensated liver disease or liver cancer. This process of progressive damage to the liver is common to both aetiologies.

While the labelling of liver disease suggests a dichotomy, the clinical reality is that there is significant overlap between ARLD and NAFLD.[4] The incidence of obesity and diabetes is rising, and a substantial proportion of the population is drinking alcohol at above recommended limits.[5]

### Strengths and limitations of this study

► This is a timely synthesis of the best available evidence on the role of moderate alcohol consumption in non-alcohol-related fatty liver disease.
► We used systematic searches to identify literature and prospectively registered our protocol on PROSPERO.
► We restricted our inclusion criteria to studies that used longitudinal data to provide evidence of temporal associations.
► Due to heterogeneity in definitions of alcohol exposure and outcomes, it was not possible to carry out a meta-analysis.
► The existing literature base is limited and only six studies were sufficiently robust to meet our predefined inclusion criteria.

**Table 1** International definitions of moderate alcohol consumption, UK recommended limits and levels that would warrant assessment for alcohol-related liver disease, all expressed in grams of alcohol and UK units

| Definitions: | Grams of alcohol | | UK units of alcohol | |
| --- | --- | --- | --- | --- |
| | Daily* | Weekly* | Daily* | Weekly* |
| Accepted International consensus of moderate alcohol consumption | **F: <20** | F: <140 | F: <2.5 | F: <17.5 |
| | **M: <30** | M: <210 | M: <3.75 | M: <26.25 |
| UK recommended safe weekly limits | ≤16 | 112 | ≤2 | **≤14** |
| NICE thresholds for assessing for liver cirrhosis | F: >40 | F: >280 | F: >5 | **F: >35** |
| | M: >57 | M: >400 | M: >7.1 | **M: >50** |

*Daily and weekly figures are given for comparison only. The bold numbering for each definition is the standard format in which this definition is expressed
NICE, National Institute of Health and Care Excellence.

It is estimated that up to 17% of the adult population may meet criteria for both NAFLD and ARLD.[6] Despite this, there is little guidance available for generalist healthcare professionals, on how to advise people with a diagnosis of NAFLD on safer alcohol consumption.

Recommendations on safe alcohol consumption levels vary worldwide. Increasingly, they take into account the effect that alcohol has on the risk of developing many adverse health outcomes, including cancer. International analysis suggests this should be as low as total abstinence to minimise all health risks.[7] Recommended limits for safe alcohol consumption in the UK general population are up to 14 units of alcohol per week in both men and women,[8] which equates to 16 g of alcohol per day at 8g/unit. Moderate alcohol consumption is generally defined in the literature as drinking within, or slightly in excess of, these limits versus complete abstinence.[4] There is a significant gap between this recommended 'moderate' limit and the levels of alcohol consumption that would prompt an assessment for alcohol-related liver damage. The UK National Institute of Health and Care Excellence (NICE) recommends offering a liver cirrhosis test to men drinking over 50 units and women drinking over 35 units a week on an ongoing basis over several months,[9] leaving a significant proportion who are drinking at and above 14 units a week, but below the levels to have liver assessment based on their alcohol consumption alone. The international differences in definition of how many grams of alcohol a 'unit' contains can create confusion and the reader is directed to table 1 to help in interpreting the study results in the context of UK Government and NICE recommended limits.

There is still uncertainty, and an absence of guidance, on safe levels of alcohol consumption for people with established NAFLD. Indeed, it is not clear that any level of alcohol consumption is safe to minimise progression of the liver disease in this population. It is known that people with very high levels of alcohol consumption (who would meet criteria for a diagnosis of ARLD), and who also have metabolic risk factors, are at even greater risk of adverse liver outcomes.[10 11] But there is also some evidence that for people with metabolic risk factors (but who do not have a NAFLD diagnosis), drinking alcohol at low levels may protect against cardiovascular disease, prevent fatty liver disease and lead to better outcomes than with complete abstinence.[12 13] Elucidating the role of alcohol in NAFLD progression is a small part of understanding the interplay of genetic and environmental factors and their effects on the liver; an area of ongoing research and debate.[14]

The purpose of this systematic review is to synthesise evidence on the role of moderate alcohol consumption on progression to severe liver disease in people with diagnosed NAFLD. This will help guide the advice given to NAFLD populations around safe alcohol consumption in primary care and specialist settings.

## METHODS
The protocol for this review was registered in advance with PROSPERO (International Prospective Register of Systematic Reviews, #CRD42020168022).

### Types of studies, inclusion and exclusion criteria
Primary studies were included if they were prospective or retrospective cohort studies. The population of interest was adult patients (>18 years old) with diagnosed NAFLD. The outcome of interest was progression of liver disease in this population. The exposure of interest was no versus moderate alcohol consumption. For our inclusion criteria we defined 'moderate consumption' as up to 35 units per week in females, and 50 units per week in males (levels that would be considered the threshold for definite risk of ARLD according to NICE guidelines[9]). This definition included studies that focused on the effects of alcohol within or just above current weekly recommended limits (the usual definition of moderate alcohol consumption), as well as those who looked beyond these levels of consumption, up to the NICE ARLD levels.

Exclusion criteria were as follows: (1) studies where the population had diagnosed ARLD; (2) studies where the population was defined according to their alcohol consumption levels rather than their NAFLD status at baseline; (3) studies where patients already had severe liver disease at the time of cohort entry; (4) cross-sectional studies or studies where exposure was only measured at the same time as outcome.

We performed a systematic review following the Preferred Reporting Items for Systematic Reviews and Meta-Analyses (PRISMA) guidelines.[15]

## Search strategy and data extraction

Potentially relevant studies were identified through systematic literature searches of relevant databases (Medline, Embase, The Cochrane Library and ClinicalTrials.gov, Conference Proceedings Citation Index—Science, Web of Knowledge, CINAHL(EBSCO)) in January 2020 and updated in September 2020. No language restrictions were applied, and databases searched documents published from 1990 onwards. Reference lists from potentially relevant papers and previous review articles were hand searched. Medical Subject Headings and free-text terms for the NAFLD population, alcohol exposures and liver outcomes of interest were used. Two researchers (HJ and either HO'K or DS) independently screened titles and abstracts. Any disagreement in full-text selection was resolved by consensus. Record screening was also assisted by Rayyan, an online software tool that assesses similarities between selected records and highlights other potentially relevant studies based on the screener's previous selection.[16] Full texts of potentially relevant papers were obtained and read by two independent researchers with reference to the predefined set of criteria to identify final study inclusion. Data were extracted into a standardised form, piloted on three studies before full extraction. Data extraction was based on the updated checklist for critical appraisal and data extraction for systematic reviews of prediction studies checklist for prognostic studies,[17] undertaken by one researcher and checked by a second. Two authors (HJ, HO'K) assessed the risk of bias independently. Since the included studies were observational cohort studies of prognostic factors, the Quality in Prognostic factor Studies tool was used.[18]

## Data synthesis

Pooling of data was not possible due to exposure and outcome heterogeneity across studies. A narrative synthesis[19] was undertaken, with data synthesised by alcohol exposure level. Due to the small number of studies, even those with high risk of bias are included in the synthesis, although this bias assessment is made clear throughout the narrative.

## Patient and public involvement

Patients and the public were not involved in the design or conduct of this review but will be involved in the dissemination of findings through a funded PPI steering group and close collaboration with the British Liver Trust.

## RESULTS

The searches identified 4578 unique citations. Of the titles and abstracts screened, 42 articles were selected for full-text screening. Thirty six were excluded at this stage for reasons summarised in the PRISMA diagram (figure 1). In seven of the excluded studies, the population did not have a baseline diagnosis of NAFLD[20–26] and in five studies the population already had advanced liver disease at baseline.[27–31] Five of the excluded studied focused on non-liver specific outcomes such as overall mortality,[32–36] while 11 were conference abstracts or short papers which held inadequate data on either population, exposure or outcomes.[20 21 23 25 29 32 33 37–40] The most common reason for exclusion at full-text stage was study design, mainly cross-sectional studies looking at a single time point to assess exposure and outcome.[24 30 31 37–39 41–47] There were also eight studies which on full-text reading were review articles or editorials.[48–55] A total of six unique studies representing data from five cohorts were eligible for inclusion in the systematic review, and were assessed for quality (figure 1).[56–61]

## Characteristics of included studies

Further details of included studies are shown in table 2.

Within the studies meeting inclusion criteria, three[58–60] looked at the exposure of alcohol consumption up to, or similar to, the accepted international definition of moderate consumption. This is <20 g/day in women and <30 g/day in men.[26] Three of the studies[56 57 61] looked at low alcohol consumption but also extended moderate consumption up to levels of alcohol consumption which would be considered more consistent with ARLD.

## Moderate alcohol consumption (accepted international definitions) and risk of liver disease progression in NAFLD

Three studies examined the effects of alcohol in NAFLD using definitions in keeping with the accepted international definition of moderate consumption.[58–60] Although these studies shared a similar aim, they varied in NAFLD population definition, measurement of alcohol consumption and choice of liver outcomes. Two looked at histological progression outcomes and one used non-invasive indirect blood-based markers of liver fibrosis. Two of the studies were rated as having a low risk of bias[59 60] and one was rated as having a moderate risk.[58]

Ajmera et al[58] studied a NAFLD population taken retrospectively from the non-alcohol related steatohepatitis (NASH) clinical research network, including populations from an observational study and the placebo arm of two NASH drug trials, all of whom had biopsy proven NAFLD (285 participants). Alcohol consumption was measured at cohort entry and at varying time points up to, and including, follow-up liver biopsy, which occurred, on average, 3.9 years later. Multiple histological markers of disease progression and resolution were studied, and the authors looked at the association between baseline drinking status and disease, as well as change in drinking status over time and disease progression/resolution. For most of the histological end points studied, there was no significant difference between moderate drinkers and abstainers in outcomes, with the only significant results suggesting that abstainers had less progressive or a higher likelihood of resolution of their disease between biopsies, particularly the persistent abstainers when compared with

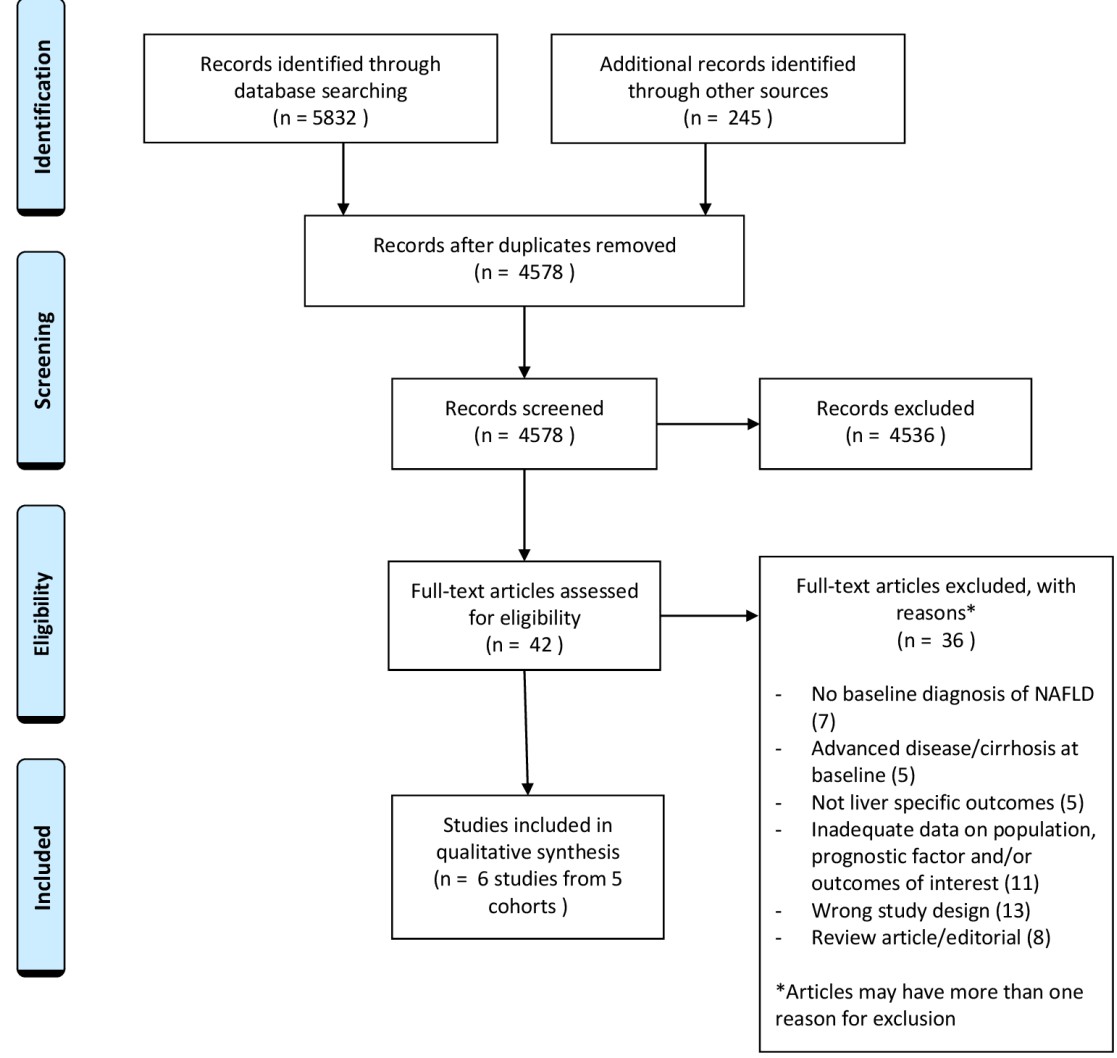

**Figure 1** Preferred Reporting Items for Systematic Reviews and Meta-Analyses diagram of study selection. NAFLD, non-alcohol-related fatty liver disease.

the persistent moderate drinkers. Results should be interpreted in the knowledge that a large number of related histological outcomes were reported, increasing the likelihood of a statistically significant result by chance. The study also had a relatively short follow-up period between biopsies. The absence of detailed information on which other prognostic factors were taken into account, led to a rating of moderate on risk of bias assessment.

A similar study by Ekstedt et al[60] looked at a smaller group (71 participants) of biopsy proven NAFLD, with follow-up histology an average of 13.8 years after initial biopsy. Alcohol consumption was assessed at baseline and follow-up, with heavy episodic drinking assessed in addition to weekly consumption. Primary outcome was significant fibrosis progression, defined as progression by one or more fibrosis stage or the development of end stage liver disease during follow-up. Although higher weekly alcohol consumption showed some tendency to predict fibrosis progression (OR for increase in grams of alcohol per week 1.012 (1.000 to 1.025)) only the presence of heavy episodic drinking (defined as >60 g/day in men and

>48 g/day in women more than once a month) reached statistical significance in predicting fibrosis progression.

Of note in both the Ajmera and Ekstedt studies were the very low levels of alcohol consumption in the 'moderate drinkers', with the majority (78%) of the moderate drinkers drinking less than monthly in the Ajmera study and the average weekly alcohol consumption in the Ekstedt study being only 39 g/week. Both studies also included a significant number of patients who already had liver inflammation (NASH) at baseline (over 50% in both studies), indicating a higher proportion of patients with a tendency to progressive disease as compared with a general NAFLD population, as would be expected with biopsy-based studies.

In contrast to the relatively selective biopsy studies, Chang et al[59] studied a large prospective population cohort (Kangbuk Samsung Health Study) of whom 58 927 had ultrasound evidence of fatty liver but without evidence of other liver diagnoses or advanced disease. Alcohol exposure was weekly units at baseline and follow-up was for a median of 8.3 years with outcome of interest being

**Table 2** Characteristics of included studies

| Author /year | Country | Study design and population | Yrs f/u | Method of NAFLD diag* | Method of measuring alcohol consumption | Definition of moderate consumption studied as RF | Study outcomes of interest and event no | Adjustments of interest considered | Adjusted HRs/OR/ mean differences for liver events with 95% CI and p values | Risk of bias |
|---|---|---|---|---|---|---|---|---|---|---|
| Åberg 2019[56] | Finland | Retrospective data linkage cohort analysis NAFLD population 6462, mean age 53 years, 60% M | 10.9 | FLI>30 | Questionnaire at cohort entry | <50 g/day in 10 g categories with abstinence as reference | Composite non-fatal and fatal liver disease 58 events | ? unclear other than age, sex | Per increase in 10 g of alcohol per day versus abstinence HR 1.43 (1.12 to 1.82) p=0.004 | High |
| Åberg et al 2020[57] | Finland (FINRISK Health survey) | Retrospective data linkage cohort analysis NAFLD population 8345, mean age 53.7 years, 60% M | 11.1 | FLI>60 | Questionnaire at cohort entry (recall for past month) | <50 g/day in 10 g categories with abstinence as reference | Composite non-fatal and fatal liver disease 152 events | Age, sex, smoking, T2DM | g alcohol/day versus abstinence 0–9 hour 1.38 (0.74 to 2.58) 10–19 hour 2.18 (1.04 to 4.53) 20–29 hour 3.62 (1.67 to 7.76) 30–39 hour 3.53 (1.53 to 8.14) 40–49 HR 8.79 (3.95 to 19.56) | Low |
| Ajmera et al 2018[58] | USA | Retrospective analysis of longitudinal cohorts within NASH CRN NAFLD population 285, mean age 47 years, 30% M | 3.9 | Liver biopsy | Questionnaire at cohort entry (Skinner lifetime drinking history) | <2 drinks per day and excluded if >6 drinks on 1 occasion ≥monthly | Histological resolution or progression on follow-up biopsy | Age, sex, race, smoking | Persistent moderate drinkers versus abstinence* **resolution** of NASH: OR 0.32 (0.11 to 0.92) p=0.04 fibrosis progression: adj mean diff 0.00 (−0.29 to 0.29) p=0.99 | Mod |
| Chang et al 2019[59] | South Korea (Kangbuk Samsung Health Study) | Prospective population cohort NAFLD population 58 927, mean age 37.7, 82% M | 4.9 | US | Questionnaire at each study visit (annual or biennial) | 10–19.9 g/day (F) 10–29.9 g/day (M) (low 1–9.9 g/day) | Fibrosis progress as estimated by high indirect serum scores** | Age, sex, BMI, smoking, exercise level, education, T2DM, BP | Mod versus abstinence† (repeat observations) Fib4: HR 1.33 (1.13 to 1.57) NFS: HR 1.37 (1.23 to 1.52) low versus abstinence (repeat observations) Fib 4: HR 1.08 (0.91 to 1.27) NFS: HR 1.14 (1.02 to 1.27) | Low |

**Table 2** Continued

| Author /year | Country | Study design and population | Yrs f/u | Method of NAFLD diag* | Method of measuring alcohol consumption | Definition of moderate consumption studied as RF | Study outcomes of interest and event no | Adjustments of interest considered | Adjusted HRs/OR/ mean differences for liver events with 95% CI and p values | Risk of bias |
|---|---|---|---|---|---|---|---|---|---|---|
| Ekstedt et al 2009[60] | Sweden | Retrospective cohort NAFLD population 71, mean age 47.3, 72% M | 13.8 | US and liver biopsy | Questionnaire AUDIT-C and interview at follow-up | g/day—no upper limit defined as 'moderate' | Fibrosis progress on follow-up biopsy no | Age, sex, BMI, T2DM, fibrosis at baseline | Increasing alcohol g/week versus abstinence OR 1.012 (1.000 to 1.025)p=0.055 | Low |
| Kawamura et al 2016[61] | Japan | Prospective cohort NAFLD population 9959, mean age 49, 87% M (included 18 patients >70g alcohol/day defined as ARLD) | 5.4 | US | Questionnaire at baseline and every 6 months | g/day in categories with <20g/day as reference | HCC on imaging | Age, sex, BMI, T2DM, serum markers | g/day alcohol versus <20g/day 20–39hour 0.90 (0.11 to 7.90) p=0.919 ≥40–69hour 2.48 (1.01 to 6.05) p=0.047 >70hour 12.61 (5.68 to 28.00) p=0.001 | Low |

*Note multiple differences in means and OR presented for different histological and biochemical outcomes between abstainers, persistent moderate drinkers, and changes in alcohol consumption between biopsies. Presented data represent histological outcomes of potential clinical prognostic significance within the remit of this review comparing persistent moderate drinking to abstinence.
†Multiple HR presented in paper for different score outcomes for single and repeated outcome measures looking at intermediate/high or high-risk scores in low and moderate drinkers and different subgroups. Presented data represent outcomes best in keeping with remit of this review using widely used indirect serum markers of liver fibrosis.
‡Scores used to estimate fibrosis progression were the Fib4 score, NAFLD fibrosis score (NFS) and AST to platelet ratio index (APRI) score.
ARLD, alcohol-related liver disease; AUDIT-C, alcohol use disorders identification test - consumption; BMI, bodymass index; BP, blood pressure; CRN, clinical research network; FINRISK, Finland cardiovascular risk study; FLI, Fatty Liver Index; g, grams; HCC, hepatocellular carcinoma; M, Male; NAFLD, non-alcohol-related fatty liver disease; NASH, non-alcohol related steatohepatitis; RF, risk factor; T2DM, type 2 diabetes mellitus; US, hepatic ultrasound; Yrs, years.

progression to advanced liver fibrosis using non-invasive blood-based markers of disease. For moderate drinkers (10–30 g/day), the risk of progressing to advanced fibrosis (using intermediate/high Fib4 score as the outcome) was HR 1.33 (1.13 to 1.57), when compared with abstainers. Light drinkers (1–10 g/day) showed a tendency towards more advanced disease when compared with abstainers, but this did not reach statistical significance (HR 1.08 CI 0.91 to 1.27).

## Moderate alcohol consumption (below the threshold that would be consistent with ARLD) and risk of liver disease progression in NAFLD

Three studies extended the definition of moderate alcohol consumption beyond the international consensus definition of moderate consumption. Two of the studies were rated as having a low risk of bias,[57 61] with one rated as high risk of bias.[56]

The general population longitudinal data presented by Chang et al[59] is supplemented by two recent related studies by Åberg et al,[56 57] using data from the same Finnish National Health Surveys (FINRISK, Health 2000) cohort. The definition of moderate alcohol consumption was increased to include anything up to 50 g/day in these studies. Although the exposures and outcome measures were the same in the two related studies, the NAFLD population was defined using different Fatty Liver Index (FLI) cut offs values, generating overlapping but distinct study populations. For this reason, data are presented from both studies.

The first study, only available as a conference abstract,[56] used a FLI>30 to retrospectively define their NAFLD population. This low FLI would generally be used as a 'rule out' rather than 'rule in' cut-off for NAFLD diagnosis[62] and the limited data presented suggests that using abstinence as a reference, any increase in alcohol consumption by 10 g/day, increased incident liver events (combined fatal and non-fatal outcomes) by 43% with a presented HR of 1.43 (1.12 to 1.82) for each 10 g rise in daily alcohol consumption. The data presented contained few details of adjustment factors or analysis plan. This study was graded as having a high risk of bias, and these results should be interpreted with caution.

A larger study,[57] based on the same cohort, retrospectively identified a NAFLD population based on a FLI of >60 (the accepted and validated cut-off for making a positive diagnosis of NAFLD in the literature[63]). Alcohol intake at cohort entry was based on estimated consumption over the previous year. Lifetime abstainers were used as the reference group. Fatal and non-fatal liver outcomes were studied in 8345 participants over 92 350 person years of follow-up. The study concluded that incident liver disease is higher at all levels of alcohol consumption, compared with lifetime abstainers with steadily rising HRs as the level of alcohol consumption increases. Although drinking up to 10 g/day was not statistically significantly different to abstaining (HR 1.38 CI 0.74 to 2.58 in the final model), levels of alcohol consumption between 10 g

and 19 g, which are roughly equivalent to the 14 units per week recommended limits, prognosticated for over double the number of incident liver events in NAFLD patients (HR 2.18 CI 1.05 to 4.53). At higher levels, which would not necessarily trigger a liver assessment for alcohol related harm in current guidelines, risk of significant liver disease was nearly nine times higher (for consumption of 40–49 g of alcohol a day, HR 8.79 CI 3.95 to 19.56).

A retrospective Japanese cohort study[61] also looked at stepwise rises in daily alcohol consumption as a prognostic factor for the more specific outcome of hepatocellular carcinoma (HCC) in people with fatty liver (identified on ultrasound). The Kawamura study with 9959 participants followed for a median of nearly 2000 days, had a reference group of people drinking <20 g of alcohol per day, rather than abstainers. This differed from all the other studies reviewed. Only those drinking at between 40 and 69 g of alcohol a day had a statistically significant increase in rates of HCC (HR 2.48 CI 1.01 to 6.05, p 0.047), with no effect in those drinking at more moderate levels. The population in this retrospective cohort were patients undergoing ultrasound at two tertiary hepatology centres in Japan rather than a general population cohort, and as HCC is known to occur in non-cirrhotic NAFLD[64] comparison with outcomes from other studies should be interpreted with caution.

Excluding the only study rated as having a high risk of bias,[56] the other good quality longitudinal studies of varying design, all reported either no association or a negative impact of moderate amounts of alcohol on future liver disease outcomes. This was seen across the studies looking at levels of alcohol consumption within the international definition of moderate consumption, and those that extended this definition of moderate consumption.

## DISCUSSION
### Summary of results
In this systematic review of the latest available longitudinal data, we found evidence to suggest that any amount of alcohol, even at low levels, may be harmful for liver health in people with diagnosed NAFLD. This evidence comes from both general population-based cohorts using coded liver outcomes, as well as tertiary centre NAFLD populations defined using histological end points.

### Comparison with existing literature
Until recently the majority of evidence in this area has come from cross-sectional studies where alcohol exposure was assessed at the same time as liver outcomes. These data provide somewhat contradictory results, with several studies indicating that moderate alcohol consumption is associated with lower levels of liver disease progression[39 43 65 66] although more recent studies support of our findings, and suggest the opposite.[42 45] The design employed in these studies does not allow the assessment of temporal relationships and is open to reverse causality

(those with liver damage may be newly abstaining from alcohol for example) in addition to recall and other biases. On the basis of these limitations, cross-sectional studies were excluded from this current review, although they have been widely cited in previous critical reviews in this area, before more recent longitudinal data were available.

In the historical absence of large prospective cohort studies and the impossibility of conducting a controlled trial in the area, comparative work has been undertaken using Mendelian randomisation. This utilises random genetic variations which affects the rate of alcohol metabolism as a proxy measure for alcohol exposure, with randomisation of patients with NAFLD based on an allele known to confer lower lifetime alcohol consumption by necessity due to the unpleasant effects of drinking even low levels of alcohol. Findings from this study were supportive of our review, with the group with higher lifetime alcohol consumption showing markers of more severe disease on biopsy, even though alcohol consumption was at very modest levels.[46]

In addition to the evidence on the relationship between modest alcohol consumption in NAFLD and liver outcomes, other published studies have focused on overall mortality and cardiovascular outcomes. A study of 4264 participants in an ultrasound diagnosed NAFLD cohort study showed no significant difference in overall mortality in those with alcohol consumption in the low/moderate range versus abstinence after 20 years of follow-up.[36] A subsequent study with the same US cohort reported a protective effect of low alcohol consumption on overall survival in NAFLD.[67] The evidence for a protective effect of low alcohol consumption on cardiovascular outcomes in the general population is generally accepted.[68] The evidence for cardiovascular protection in those with NAFLD is more limited, with some evidence that moderate alcohol may provide some benefit[69] but more recent studies finding no protective effects.[42 70] The comparative evidence on overall mortality and cardiovascular outcomes highlights the need to assess liver disease risks within these competing contexts.

## Strengths and limitations
Although there have been several recent critical reviews of the role of moderate alcohol consumption in NAFLD, the most recent of which reach similar conclusions,[4 49 71] these have been wider in their remit with less well-defined inclusion criteria and less systematic methodology. The predetermined inclusion criteria, robust systematic data collection and reporting techniques (in line with PRISMA guidelines) and decision to avoid cross-sectional data are all important in providing the best available evidence to answer the review question of the temporal relationship between moderate alcohol consumption and liver outcomes in NAFLD. The challenges of synthesising observational data, including unmeasured confounding and heterogeneity, were anticipated, but meant that data pooling was not possible.

A particular limitation hindering comparison between studies was the methods of defining moderate alcohol consumption. The consensus for defining a level of alcohol consumption above which a diagnosis of pure NAFLD cannot be made have been supported by the European Association for the Study of the Liver and the American Association for the Study of the Liver Diseases and set at 20 g/day in women and 30 g/day in men,[62 72] yet most of the published studies do not use these cut-offs in their data. Until this is standardised across studies, with an additional consensus defining levels above this moderate but not high enough to reach levels associated with a definite diagnosis of ARLD, synthesising the evidence in this area will remain challenging.

## Implications for research/practice
This review adds weight to individual studies showing that any level of alcohol intake in NAFLD may be harmful to liver health. Further prospective cohort studies are needed, with detailed definitions/measures of alcohol exposure, and validated clinical liver outcomes, measured at appropriate times. Future research should focus on looking at outcomes in relation to accepted alcohol intake levels used in definitions of NAFLD. It should also take into account that the clinical reality is a dual-aetiology patient who may currently be excluded from both diagnostic categories based on their alcohol intake being too high for NAFLD, and too low for ARLD definitions. This is an ever-expanding patient group seen in many clinical settings.

Based on a synthesis of the evidence presented in this review, clinicians seeing patients with NAFLD in primary or secondary care should currently advise abstinence from alcohol to avoid accelerating liver harm. This is likely to be difficult for patients to accept, and public health messaging will need careful thought if it is to have any impact on liver health.

**Contributors** HJ: conceptualisation, methodology, conduct, analysis, writing of initial draft, guarantor. HO'K: methodology, conduct, figures/infographic, critical review and comments on drafts. DC: supervision of methodology, conduct, critical review and comments on drafts. DS: methodology, conduct, critical review and comments on drafts. BH: conceptualisation, supervision of methodology, conduct, analysis, critical review andcomments on drafts. QMA: conceptualisation, supervision of conduct and analysis, critical review and comments on drafts.

**Funding** This work was supported by and NIHR Doctoral Research Fellowship – HJ personal award. Award no: NIHR300716.

**Competing interests** HJ reports grants from National Institute for Health research (NIHR), during the conduct of the study; personal fees from Intercept Pharma, personal fees from Norgine, outside the submitted work; QMA reports grants from European Commission, during the conduct of the study; other from Acuitas Medical, grants, personal fees and other from Allergan/Tobira, other from E3Bio, other from Eli Lilly & Company Ltd, other from Galmed, grants, personal fees and other from Genfit SA, personal fees and other from Gilead, other from Grunthal, other from Imperial Innovations, grants and other from Intercept Pharma Europe Ltd, other from Inventiva, other from Janssen, personal fees from Kenes, other from MedImmune, other from NewGene, grants and other from Pfizer Ltd, other from Raptor Pharma, grants from GlaxoSmithKline, grants and other from Novartis Pharma AG, grants from Abbvie, personal fees from BMS, grants from GSK, other from NGMBio, other from Madrigal, other from Servier, outside the submitted work.

**Patient consent for publication** Not applicable.

**Ethics approval** This study does not involve human participants.

**ORCID iDs**
Helen Jarvis http://orcid.org/0000-0001-5039-0228
Daniel Stow http://orcid.org/0000-0002-9534-4521
Barbara Hanratty http://orcid.org/0000-0002-3122-7190

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
