## [Reviewer comments · BMJ Open]

ARTICLE DETAILS

TITLE (PROVISIONAL)	Does moderate alcohol consumption accelerate the progression of liver disease in NAFLD? A systematic review and narrative synthesis
AUTHORS	Jarvis, Helen; O'Keefe, Hannah; Craig, Dawn; Stow, Daniel; Hanratty, Barbara; Anstee, Quentin

VERSION 1 – REVIEW

REVIEWER	Tarantino, Giovanni Federico II University Medical School, ClinandExpert Medicine
REVIEW RETURNED	19-Mar-2021

GENERAL COMMENTS	Authors to give a more balanced view of the topic should comment at large on the following papers whose results are likely in contrast with presented data. Suzuki A, et al. Light to moderate alcohol consumption is associated with lower frequency of hypertransaminasemia. Am. J. Gastroenterol. 2007;102:1912–1919. Dunn W, Xu R, Schwimmer JB. Modest wine drinking and decreased prevalence of suspected nonalcoholic fatty liver disease. Hepatology. 2008;47:1947–1954. Gunji T, et al. Light and moderate alcohol consumption significantly reduces the prevalence of fatty liver in the Japanese male population. Am. J. Gastroenterol. 2009;104:2189–2195. Moriya A, et al. Alcohol consumption appears to protect against non-alcoholic fatty liver disease. Aliment. Pharmacol. Ther. 2011;33:378–388. Furthermore, authors should stress that the mechanisms underlying NAFLD, although many efforts , are far from being discovered, reason for which the the therapeutical approaches are at the best mediocre, as clearly emphasized by.... J. Clin.Med.2020,9(1),15;https://doi.org/10.3390/jcm9010015.
--

REVIEWER	Olivier, Jake University of New South Wales, School of Mathematics and Statistics
REVIEW RETURNED	26-Jul-2021

GENERAL COMMENTS	In this systematic review of non-alcohol related fatty liver disease (NAFLD), the authors identified six studies that met inclusion criteria and decided not to synthesise the numerical summaries of these studies into a meta-analysis. The decision to not conduct a meta-analysis does not appear unreasonable; however, this is not clear until reading 3/4 or more of the manuscript.
--

	Please make it clear why a meta-analysis was not conducted and justify this decision in the abstract. The abstract solely states it was not done due to "heterogeneity". The term heterogeneity can take on many different meanings and it is not clear which meaning is being used here. From what I can gather, the studies differ with regards to: (1) method of NAFLD diagnosis, (2) study outcomes of interest, and (3) whether the study design allowed for computing the hazard ratio or odds ratio. Are these all the reasons for between study heterogeneity? The abstract/introduction should include more information about how liver disease is diagnosed, measured and tracked over time. Much of this information is provided far too late in the manuscript in the results/discussion sections. The intro should also include a discussion about what is meant by a "unit of alcohol". This is not a standard definition worldwide, from what I understand about the topic. This alone can make it difficult to synthesise studies across different countries that have different definitions. Page 6. It is not clear why defining moderate alcohol consumption is an inclusion criterion. This is how groups have been defined and not how studies were selected. The conclusions regarding alcohol intake at any level do not appear supported by the data, especially the advice to abstain from alcohol. There seems to be a clear message of harm for moderate and heavy drinking, but not for lower levels of alcohol intake as per the results of the included studies. In fact, the authors cite studies where low level of alcohol consumption was beneficial. Are the data used in Adberg et al (2019) a subset of the data used in Adberg et al (2020)? If so, then this creates problems with dependence and, effectively, double counting. It could be argued the less complete study, presumably the abstract only document, should be excluded. Other issues: Page 4. What does "<65" mean? Is this an age range? Page 6. The wording of "databases were searched from 1990 onwards" implies the researchers started this project in year 1990. Presumably, this is meant to be "searched documents published from 1990 onwards". Reviewed studies that have been excluded with reasons should contain citations to provide the reader an understanding of why each study was excluded. Table 2 is not a study result and should have been known by the researchers prior to this review. Discussion: Criticisms of cross-sectional studies is reasonable, but I question whether it is reasonable to argue the knowledge gained from the included studies is a clear improvement given the heterogeneity between studies.
--	--

	Page 14, last paragraph. Please cite this study in the first sentence. I erroneously believed this was additional work taken on by the current study authors.
--	---

VERSION 1 – AUTHOR RESPONSE

Reviewer 1 comments:

Authors to give a more balanced view of the topic should comment at large on the following papers whose results are likely in contrast with presented data. (papers then listed)

Suzuki A, et al. Light to moderate alcohol consumption is associated with lower frequency of hypertransaminasemia. *Am. J. Gastroenterol.* 2007;102:1912–1919.

Dunn W, Xu R, Schwimmer JB. Modest wine drinking and decreased prevalence of suspected nonalcoholic fatty liver disease. *Hepatology.* 2008;47:1947–1954.

Gunji T, et al. Light and moderate alcohol consumption significantly reduces the prevalence of fatty liver in the Japanese male population. *Am. J. Gastroenterol.* 2009;104:2189–2195.

Moriya A, et al. Alcohol consumption appears to protect against non-alcoholic fatty liver disease. *Aliment. Pharmacol. Ther.* 2011;33:378–388.

The papers listed are all familiar to the authors. They did not meet the inclusion criteria for this systematic review as the outcomes do not relate to progression of liver disease, but rather to the prevalence of NAFLD or elevated liver blood tests (presumed to be due to NAFLD) in people with varying levels of alcohol consumption. They look to answer a different question around the role of low/moderate alcohol consumption on the risk of developing NAFLD in a population without the condition.

In the introduction in the third paragraph we mention and highlight this evidence:

‘there is also some evidence that for people with metabolic risk factors (but who do not have a NAFLD diagnosis), drinking alcohol at low levels may protect against cardiovascular disease, prevent fatty liver disease, and lead to better outcomes than with complete abstinence (12,13).’

The reference given (13 - Sookoian S, Castaño GO, Pirola CJ. Modest alcohol consumption decreases the risk of non-alcoholic fatty liver disease: a meta-analysis of 43 175 individuals. *Gut.* 2014 Mar;63(3):530–2.) is a meta-analysis on the role of moderate alcohol consumption and the risk of developing NAFLD which includes data on 3 of the 4 references suggested by reviewer 1. The authors feel rather than providing a more balanced view on the same topic this is a distinct question and hence our positioning of it in the introduction rather than in the ‘comparison with existing literature’ section of the discussion.

Furthermore, authors should stress that the mechanisms underlying NAFLD, although many efforts , are far from being discovered, reason for which the the therapeutical approaches are at the best mediocre, as clearly emphasized by.... *J. Clin.Med.*2020,9(1),15;https://doi.org/10.3390/jcm9010015.

Thank you for raising this important point. We have added a line to the introduction and added this reference :

'Elucidating the role of alcohol in NAFLD progression is a small part of understanding the interplay of genetic and environmental factors and their effects on the liver; an area of ongoing research and debate (14).'

Reviewer 2 comments:

Please make it clear why a meta-analysis was not conducted and justify this decision in the abstract. The abstract solely states it was not done due to "heterogeneity". The term heterogeneity can take on many different meanings and it is not clear which meaning is being used here. From what I can gather, the studies differ with regards to: (1) method of NAFLD diagnosis, (2) study outcomes of interest, and (3) whether the study design allowed for computing the hazard ratio or odds ratio. Are these all the reasons for between study heterogeneity?

Thank you for clearly wording the authors' dilemma. We agree that the primary reason for not being able to meta-analyse the data was difference in the definition of the exposure of 'moderate consumption of alcohol'. As is pointed out later on by reviewer 2, there is no international consensus as to how many grams of alcohol define a unit, nor the number of units deemed to be at or below recommended guidelines. The studies we included in our synthesis were, accordingly, using different cut-offs to group participants. Differences in outcome measures used would also make pooled analysis difficult (or inappropriate). The difference in effect measures/ model results could potentially have been accounted for (depending on data availability), and hence were not detailed in the abstract. The abstract conclusion now reads:

'Study heterogeneity in definitions of alcohol exposure as well as in outcomes limited quantitative pooling of results. Use of standardised definitions for exposure and outcomes would support future meta-analysis.'

In view of the limited word count in the structured abstract we believe we have sufficiently justified the decision not to meta-analyse study results.

The abstract/introduction should include more information about how liver disease is diagnosed, measured and tracked over time. Much of this information is provided far too late in the manuscript in the results/discussion sections.

Thank you for this comment, which we think is important for the non-specialist reader. We have added additional sentences to the introduction (see below) following your recommendation.

'Chronic liver disease is often diagnosed as a result of abnormal liver blood tests or liver imaging, with a fatty liver (steatosis) progressing in some through inflammation (steatohepatitis) and stiffening (fibrosis) to scarring (cirrhosis) increasing the risk of decompensated liver disease or liver cancer. This process of progressive damage to the liver is common to both aetiologies.'

The intro should also include a discussion about what is meant by a "unit of alcohol". This is not a standard definition worldwide, from what I understand about the topic. This alone can make it difficult to synthesise studies across different countries that have different definitions.

In response to this comment and a later comment regarding Table 2 not being part of the study results we have moved what was Table 2 (now Table 1) to the introduction. We have also added a sentence before the table appears which reads:

The international differences in definition of how many grams of alcohol a 'unit' contains can create confusion and the reader is directed to table 1 to help in interpreting the study results in the context of UK Government and NICE recommended limits.

Page 6. It is not clear why defining moderate alcohol consumption is an inclusion criterion. This is how groups have been defined and not how studies were selected.

As the exposure of interest was 'moderate alcohol consumption' we felt that we needed to have a broad definition of this within the inclusion criteria. Studies were selected on the basis that they included the exposure by the definition given. This was to avoid selecting studies reporting on high/hazardous levels of alcohol consumption in NAFLD which is known to be harmful (as for the general population).

The conclusions regarding alcohol intake at any level do not appear supported by the data, especially the advice to abstain from alcohol. There seems to be a clear message of harm for moderate and heavy drinking, but not for lower levels of alcohol intake as per the results of the included studies. In fact, the authors cite studies where low level of alcohol consumption was beneficial.

None of the studies meeting the inclusion criteria and included in the narrative synthesis showed low levels of alcohol to be beneficial. The majority studying the effects even at what would be considered moderate consumption showed evidence of harm with some showing no effect or not reaching statistical significance. In synthesising the results the authors felt the evidence available could not be used to recommend that any level of alcohol was beneficial and therefore the cautious conclusion was to recommend abstinence with current best evidence.

Are the data used in Adberg et al (2019) a subset of the data used in Adberg et al (2020)? If so, then this creates problems with dependence and, effectively, double counting. It could be argued the less complete study, presumably the abstract only document, should be excluded.

Thank you for this comment. The data used in Adberg (2019) is from a different population as they used a different definition of NAFLD with different cut-off points, although it is acknowledged that data comes from the same wider cohort. Details of how overlapping the study cohorts actually are is lacking due to the first study (Adberg (2019)) only being available as a conference abstract. The authors felt the issue of double counting was less relevant as no quantitative synthesis was being attempted and we clearly specify there is a high risk of bias with this study and results should be interpreted with caution. We also specify in the summarising paragraph of the results section that our conclusions exclude taking into account this study due to the high risk of bias. We still feel it provides some additional information for the reader in this field and would prefer to include this study with these caveats as clearly stated.

Page 4. What does "<65)" mean? Is this an age range?

Yes - have now altered this to (<65 yrs) to make this clearer

Page 6. The wording of "databases were searched from 1990 onwards" implies the researchers started this project in year 1990. Presumably, this is meant to be "searched documents published from 1990 onwards".

Thanks for pointing this out. I have altered the text accordingly to what you suggest.

Reviewed studies that have been excluded with reasons should contain citations to provide the reader an understanding of why each study was excluded.

The reasons for study exclusion are detailed in the PRISMA diagram. Providing citations for all the studies excluded, even at full text stage, would mean an additional 36 references. The authors would be happy to provide these references if felt relevant by the editor but feel that a summary of exclusion reasons should provide sufficient information for the majority of the readers.

Table 2 is not a study result and should have been known by the researchers prior to this review.

Per your recommendation, we have moved this table into the introduction section and renamed as Table 1.

Discussion: Criticisms of cross-sectional studies is reasonable, but I question whether it is reasonable to argue the knowledge gained from the included studies is a clear improvement given the heterogeneity between studies.

The authors feel that in view of the included studies including temporal associations between exposure and outcomes, that this is a clear improvement to the quality of the evidence from which to draw conclusion. We have acknowledged the heterogeneity of these studies as a limitation in the discussion and have used it as a basis for clear recommendations for future research.

Page 14, last paragraph. Please cite this study in the first sentence. I erroneously believed this was additional work taken on by the current study authors.

Thank you for pointing this out: the citation has been moved to early in the first sentence.

VERSION 2 – REVIEW

REVIEWER	Olivier, Jake University of New South Wales, School of Mathematics and Statistics
REVIEW RETURNED	29-Oct-2021

GENERAL COMMENTS	Thanks for addressing my previous comments. The summary for Ajmera (2018) in Table 1 includes: "OR of outcome persistent moderate drinkers v abstinence** resolution of NASH OR 0.32 (0.11-0.92) p0.04 fibrosis progression OR 0.00 (-0.29-0.29) p0.99" Considering the other HR/OR are in the direction of >1 if alcohol use were associated with a negative outcome, it would appear to be a positive outcome here. At a minimum this result is confusing given the authors' conclusions. Note that it is not possible for an OR or HR to have negative values and presumably you mean p=0.04 and similarly for other p-values. Kawamura (2016) is also problematic since they report HR=0.90, which is in the direction of being beneficial. The p-value is large
--

	but that does not coincide with a conclusion that any alcohol use has a negative impact. Re not citing excluded studies: There needs to be transparency in the decisions made to include and exclude studies. This includes not just providing reasons studies have been excluded but linking those reasons with the studies. Otherwise, you are hiding critical decisions that can influence your conclusions. Note the PRISMA statement states:  - Give numbers of studies screened, assessed for eligibility, and included in the review, with reasons for exclusions at each stage, ideally with a flow diagram. Authors usually accomplish this by providing citations and reasons for excluding studies. Also note that BMJ Open does not have a page limit, so another 36 citations is not an issue.
--	--

VERSION 2 – AUTHOR RESPONSE

Reviewer: 2

Dr. Jake Olivier, University of New South Wales

Comments to the Author:

Thanks for addressing my previous comments.

The summary for Ajmera (2018) in Table 1 includes:

"OR of outcome persistent moderate drinkers v abstinence** resolution of NASH OR 0.32 (0.11-0.92)
p0.04

fibrosis progression OR 0.00 (-0.29-0.29) p0.99"

Considering the other HR/OR are in the direction of >1 if alcohol use were associated with a negative outcome, it would appear to be a positive outcome here. At a minimum this result is confusing given the authors' conclusions. Note that it is not possible for an OR or HR to have negative values and presumably you mean p=0.04 and similarly for other p-values.

Thank you for pointing out this lack of clarity in the table. The first outcome is resolution of NASH so the OR of 0.32 presented represents the result that persistent moderate drinkers had less resolution of their liver disease than those who were persistently abstinent i.e. this is a negative outcome of drinking. We agree that this is a slightly confusing way of presenting this data but is as presented in the original paper. We have tried to make this clearer in the table by adding the word resolution in

bold and also added this to the outcome of interest column. We have made a few small changes to the narrative section relating to the Ajmera paper which we hope also makes this clearer.

The second outcome of liver fibrosis progression should have stated that the outcome measure was adjusted mean difference rather than an OR. Thank you for pointing this out. We have now made this clear and added mean difference both in this box and also to the title of the relevant column. We have added = to all the available p values in the table - thank you for pointing this out.

Kawamura (2016) is also problematic since they report HR=0.90, which is in the direction of being beneficial. The p-value is large but that does not coincide with a conclusion that any alcohol use has a negative impact.

Thank you for pointing this out. As explained in the narrative section pertaining to this study, this study differed from all the others in not using abstinence as the reference group, but rather those drinking < 20 g of alcohol making comparison difficult as we are not given information of how this reference group compares to abstinence. Although this study suggests that this level of alcohol may not be harmful for the specific outcome of hepatocellular carcinoma (HCC) compared to lower levels of alcohol, this did not reach statistical significance with a large confidence interval and P=0.919.

We are careful in the conclusions to use the term 'may be harmful' as this is the direction of the majority of the evidence we present. Due to this clear direction of the evidence and a lack of any statistically significant evidence showing a protective effect, the conclusion to recommend abstinence as the 'safest for health' option in people with NAFLD is, we feel, justified.

As a result of this comment we have adjusted the final conclusion to make it clear that we are basing the recommendation on an overall synthesis of the evidence presented.

Re not citing excluded studies: There needs to be transparency in the decisions made to include and exclude studies. This includes not just providing reasons studies have been excluded but linking those reasons with the studies. Otherwise, you are hiding critical decisions that can influence your conclusions.

Note the PRISMA statement states:

- Give numbers of studies screened, assessed for eligibility, and included in the review, with reasons for exclusions at each stage, ideally with a flow diagram.

Authors usually accomplish this by providing citations and reasons for excluding studies. Also note that BMJ Open does not have a page limit, so another 36 citations is not an issue.

Thank you for raising this. We have adjusted the PRISMA flow diagram so that the box detailing reasons for exclusion at full text stage states the numbers of studies excluded for each reason given. We have also added a short section to the first paragraph of the results text section to give more detail and reference to the studies excluded at the full text screening stage.